

# Iron oxides in the cryoconite on the glaciers over Tibetan Plateau: abundance, speciation and implications

Zhiyuan Cong[1, 5, *], Shaopeng Gao[1], Wancang Zhao[3], Xin Wang[4], Guangming Wu[1, 6],

Yulan Zhang[2], Shichang Kang[2, 5, *], Yongqin Liu[1], and Junfeng Ji[3]

[1]Key Laboratory of Tibetan Environment Changes and Land Surface Processes, Institute of Tibetan Plateau Research, Chinese Academy of Sciences, Beijing 100101, China
[2]State Key Laboratory of Cryospheric Sciences, Northwest Institute of Eco-Environment and Resources, Chinese Academy of Sciences, Lanzhou 730000, China
[3]Key Laboratory of Surficial Geochemistry, Ministry of Education, School of Earth Sciences and Engineering, Nanjing University, Nanjing 210023, China
[4]Key Laboratory for Semi-Arid Climate Change of the Ministry of Education, College of Atmospheric Sciences, Lanzhou University, Lanzhou 730000, China
[5]CAS Center for Excellence in Tibetan Plateau Earth Sciences, Beijing 100101, China
[6]University of Chinese Academy of Sciences, Beijing 100049, China

*Correspondence to*: Zhiyuan Cong (zhiyuancong@itpcas.ac.cn) or Shichang Kang (shichang.kang@lzb.ac.cn)

**Abstract.** As an important constituent of light-absorbing impurities on the glacier surface, iron oxides determine the radiative impact of mineral dust. In particular, the distinct optical properties between hematite and goethite highlight the necessity to obtain accurate knowledge about their abundance and speciation. Cryoconite samples from five glaciers in different region of Tibetan Plateau (TP) and surroundings were studied. The iron abundances in the cryoconite from TP glaciers ranged from 3.40% to 4.90% by mass, which reflected a natural background level. Iron oxides were extracted and determined using diffuse reflectance spectroscopy. The ratios of free to total iron for the five glaciers ranged from 0.31 to 0.70, emphasizing that iron in the form of oxides should be considered rather than total iron in the albedo and radiative modelling. Furthermore, the goethite content in iron oxides (in mass fraction) ranged from 81% to 98%, showing that goethite is the predominant form among the glaciers. Using the abundance and speciation of iron oxides as well as their optical properties, the total light absorption were quantitatively attributed to goethite, hematite, black carbon and organic matters at 450 nm and 600 nm wavelengths. We found that the goethite played a stronger role than BC at shorter wavelength for most glaciers. Such findings are essential to understand the relative significance of anthropogenic/natural effect, and then taking the proper mitigation measures.

## 1 Introduction

The light-absorbing impurities (LAIs) in glaciers could significantly reduce the surface albedo of snowpack and absorb more solar energy (Warren and Wiscombe, 1985). LAIs were recognized as a major contributor to the glacier and ice sheet melting (Qian et al., 2015), along with the rising air temperatures (IPCC, 2014). The composition of LAIs on the glacier surface is very complex. Their major constituents include black carbon, brown carbon, soil dust, as well as biogenic matter (Baccolo



et al., 2017; Kaspari et al., 2015; Pu et al., 2017; Takeuchi, 2002). The complex of LAIs is particularly the case for the mountain glaciers in Himalayas and Tibetan Plateau (TP) (Fig. 1). Their surfaces are commonly covered by incoherent impurities (granular sediment) with dark color, which was termed as cryoconite (Baccolo et al., 2017; Dong et al., 2016).

Presently, tremendous attention has been paid to the black carbon (Kaspari et al., 2011; Wang et al., 2015; Yasunari et al., 2010), partly because the cryosphere region of TP is in the vicinity of intensive BC source region, e.g. South Asia, and is receiving strong influences from those anthropogenic emissions (Cong et al., 2015a). Nevertheless, few research focus on dust in this region despite the observation that dust is apparently the predominant constituent of impurities on the glacier surface,

especially in its ablation area (Qian et al., 2015). Based on field research at Mera Glacier of Nepal Himalayas, Kaspari et al. (2014) pointed out that when dust loading is high, the snow albedo reduction and subsequent radiative forcing caused by dust will overwhelm black carbon.

To quantify the relative contribution of dust and black carbon as well as other substances is challenging (Painter et al., 2010). In the snow albedo simulation model like SNICAR (Flanner and

Zender, 2006), dust concentration (micrograms of dust per gram of ice) was employed to represent the loading of dust, without considering the dust composition. Actually, the light-absorption capacity of dust essentially depends on the presence of iron oxides (also commonly termed as "free iron") (Alfaro et al., 2004; Moosmüller et al., 2012; Shi et al., 2012). The most common iron oxide species in nature are hematite ($Fe_2O_3$) and goethite ($FeO(OH)$), which have distinct optical properties in terms of

absorption and wavelength dependence (Balsam et al., 2014). At the same time, some parts of iron are incorporated into the crystal lattice of alumino-silicates (defined as structural iron), and they do not contribute to the absorption of solar light (Lafon et al., 2006).

Up to now, the iron abundance and especially its mineral phases in glacial area are not well understood (Shahgedanova et al., 2013). The degree to which iron oxides contribute to solar absorption

and reduction of snow albedo remain uncertain. To constrain the uncertainties of estimating the radiative forcing of cryoconite, particularly in the Tibetan Plateau glacier area, we designed this research to address several key issues. Firstly, what is the abundance of iron in the cryoconite of mountain glaciers? How much fraction of the total iron exists as iron oxides with efficient light-absorption capability? What is the relatively proportion of hematite and goethite, considering their

distinct optical characteristics? What is their spatial variation in different types of glacier? Furthermore, how do the iron oxides impact the absorbing properties of cryoconite?

## 2 Field sampling and laboratory measurements

### 2.1 Field sampling

In order to examine geographic variability, five glaciers in different region of TP and surroundings

were chosen for the cryoconite sampling (Fig. 2). The detailed descriptions on the location and elevation are summarized in Table 1. Urumqi No. 1 Glacier (43°06′N, 86°49′E) (hereafter donated as UG), with two branches covering 1.646 km², is located in eastern Tien Shan. The air circulation regime



there is dominated by the westerlies in summer and the Siberian High in winter (Wang et al., 2014). Laohugou Glacier (donated as LHG), with a length of 10 km and an area of 20 km$^2$, is at the northern slope of western Qilian Mountains with typical continental climatic conditions (Dong et al., 2014). That area is surrounded by large sandy deserts in Northwest China, like Taklimakan Desert to the west,

Qaidam Basin to the southwest and the Gobi Desert to the north. Xiaodongkemadi Glacier (donated as XDK) is located on the northern slope of Tanggula Mountains, center of Tibetan Plateau. Previous studies showed that Tanggula Mountain is the north most boundary of the South Asian monsoon influence. Palong #4 Glacier (donated as PL), located in the southeast Tibetan Plateau, is a typical temperate glacier. It is strongly influenced by the South Asian summer monsoon intruding via the

Brahmaputra Valley, and characterized by high accumulation (2500-3000 mm) and ablation rates on an annual scale (Yang et al., 2015). Baishui No. 1 Glacier (donated as BS), with length 2.26 km and area 1.32 km$^2$, is the largest glacier in Yulong Mountains, southeast margin of Tibetan Plateau. It is characterized by high precipitation, low snow line, and relatively high temperatures (equilibrium line mean annual value -6 °C, summer value 1-5 °C) (Niu et al., 2013).

Cryoconite samples were collected using a stainless-steel scoop on the surfaces of five glaciers described above. Samples were preserved in NALGENE HDPE wide-mouth bottles (500 ml) and kept in frozen until analysis. In the laboratory the cryoconite samples were freeze-dried, turning into powder for the subsequent determination. Therefore, in this work, the cryoconite refers to the dried mass of cryoconite, excluding the water (snow or ice) contents.

**2.2 Elemental analysis by ICP-MS**

A portion of cryoconite sample was digested in a laboratory hood with 1% HNO$_3$ (Optima Grade, Fisher Co.), then measured by inductively coupled plasma-mass spectrometry (ICP-MS, Thermo X7, Thermo-Elemental Corp.) for Fe and other elements. The accuracy and precision of trace elements was ascertained based on repeated measurement of the USGS Geochemical Reference Standard (Andesite,

AGV-2). The measured and certified values for Fe agree well, with recovery better than 95%. The detailed description of the analytical protocol could be found in previous work (Cong et al., 2015b; Wu et al., 2009).

**2.3 Total organic carbon and black carbon**

The contents of organic carbon were determined by a total carbon analyzer (TOC-V, Shimadzu). The

accuracy of the TOC analysis was ±5%. The separation and analysis of black carbon in the cryoconite were adopted from the procedures previously developed for sediments (Cong et al., 2013; Han et al., 2011). Specifically, the samples were first freeze-dried, grinded into powder and weighed. Then, HCl (2N) was added, to remove carbonates, silicates, and some kinds of metal oxides. The solution was centrifuged to remove the supernatants. Then mixture (1:2) of HCl (6N) and HF (48%) were added into

the residue and reacted further. Finally, the residual solid was diluted with ultrapure water, filtered by quartz fiber filter with even distribution on their surface (QMA grade; Whatman International Ltd, England). The quartz filters were analyzed for BC using a DRI model 2001 carbon analyzer. For



quality control, standard reference material (marine sediment, NIST SRM-1941b) was also analyzed (Cong et al., 2013).

## 2.4 Extraction and quantification of iron oxides

Cryoconite samples were treated with Citrate-Bicarbonate-Dithionite (CBD) method three times to completely extract iron oxides (Ji et al., 2002; Lafon et al., 2004; Mehra and Jackson, 1958). Then dissolved $Fe^{3+}$ concentrations in the CBD solution were determined by a UV-2100 spectrophotometer (UNICO Inc., Shanghai) to obtain the iron mass in the form of oxides, Fe(ox), relevant to the light-absorption in the visible light. The remaining iron, i.e. the structural iron, was calculated by subtracting the Fe(ox) from total iron.

Fe(struc) = Fe(tot) – Fe(ox)

Here, Fe(tot) is the total iron concentration achieved from ICP-MS elemental analysis.

## 2.5 Hematite/goethite measurement using Diffuse Reflectance Spectroscopy

Given the low abundance of Fe in the cryoconite, the speciation of iron oxides can not be achieved by traditional mineralogical analysis methods like X-ray diffraction (XRD). In this study, diffuse reflectance spectroscopy (DRS) was employed to distinguish and quantify hematite and goethite. Measurements were conducted using a Perkin-Elmer lambda 900 spectrophotometer (Perkin-Elmer Corp., Norwalk, CT) equipped with a diffuse reflectance attachment. Analyses were performed for spectra in the range from 400 to 700 nm with an interval of 2 nm. Detailed procedures have been well described previously (Ji et al., 2002; Lu et al., 2017).

A set of calibration samples containing known hematite were measured. Then the percent reflectance in red color band (630-700 nm, redness) was used as an independent variable in a transfer function for calculating hematite, which was established through regression as following:

$Hm(wt. \%) = 1E\text{-}06 \cdot e^{27.37*Redness}$ ($R^2_{adj}$ = 0.9301, RMSE = 0.3018)

Assuming the CBD-extracted Fe (in the form of iron oxides) are only constituted by hematite (Hm) and goethite (Gt), the content of goethite could be calculated using the following equation:

Gt (wt. %) = 1.59 × ($Fe_{(ox)}$-Hm/1.43)

The reproducibility standard deviation of reflectance at all wavelengths was less than 0.15% (Lu et al., 2017).

## 2.6 Light absorption of cryoconite

Measurements of light absorption were performed using an ISSW spectrophotometer in Lanzhou University, China. The experimental strategy was mainly based on the method described by Doherty et al. (2010) and Wang et al. (2013). The ISSW measurement system are specially designed to be sensitive to light absorption and to avoid the interference of light scattering (Grenfell et al., 2011). The



ISSW spectrophotometer could provide the spectral absorption properties of cryoconite, by weighting the transmitted light ($I$) for a sample and that for a blank filter ($I_0$). The relative attenuation ($x_\lambda$) described by the natural logarithm of $I_0 / I$:

$$x_\lambda = \ln[\,I_0(\lambda)/I(\lambda)\,] \qquad\qquad (1)$$

5     The spectrum of light attenuation was further calibrated by a set of black carbon standards (fullerene soot, Alfa Aesar, Inc., Ward Hill, MA, USA) (Fig. S1). The light attenuation by samples on filter at specific wavelength will be converted to equivalent BC mass loading ($L_{BC}$, μg C cm$^{-2}$), which allows to calculate absorption optical depth $\tau(\lambda)$ by cryoconite: $\tau_\lambda = L_{BC}\beta_\lambda$. Where $\beta_\lambda$ is mass absorption coefficiency (MAC) of standard black carbon (i.e., fullerene, 6.3 m$^2$ g$^{-1}$ at 550 nm) (Grenfell et al., 2011; Zhou et al., 2017). Then the light absorption capacity of cryoconite was calculated through dividing the absorption optical depth by the mass loading on filter:

$$MAC = \tau_\lambda / L \qquad\qquad (2)$$

The absorption Ångström exponent (AAE) describes the wavelength dependence of the light absorption by particles (Ångström, 1929). The value of AAE could obtained by the formula of: AAE = $-\ln(\tau_1/\tau_2)/\ln(\lambda_1/\lambda_2)$, where $\tau_1$ and $\tau_2$ are the light attenuation calculated at given wavelength $\lambda_1$ and $\lambda_2$, respectively.

## 3 Results and discussion

### 3.1 Organic carbon and black carbon contents

The total organic carbon and black carbon mass fractions of the cryoconite from the five glaciers are presented in Figure 3. The most striking feature was that BS exhibited the highest TOC content (9.70 ± 0.99 % in mass faction), about 4 times higher than other four glaciers. Similarly, the BC in the cryoconite from BS were also significantly higher than other glaciers, i.e. UG, LHG, XDK and PL. For the black carbon concentration, BS glacier also has the highest abundance (1.99 ± 0.28% in mass), indicating the strong anthropogenic (fossil fuel and biomass burning) influence there. For the remaining four glaciers, their black carbon contents were comparable, ranging from 0.06 ± 0.01% (in total mass of dried cryoconite) of XDK to 0.13 ± 0.03% of PL. For comparison, Di Mauro et al. (2017) reported the black carbon values in cryoconites from Morteratsch Glacier (Swiss Alps), with the range of 0.30 - 0.4 % in mass faction.

### 3.2 Abundance of elemental Fe (total) and free-Fe (iron oxides)

The iron contents found in cryoconite samples from UG, LHG, XDK, PL and BS glaciers averaged 4.62%, 4.28%, 3.40%, 4.18% and 4.90%, by mass, respectively (Figure 4). Our data were similar to the previous reported iron contents in dust particles preserved in ice cores across Tibetan Plateau (Wu et al., 2012), which ranged from 3.38% to 5.41%. The iron in the cryoconite on the TP glaciers represents a natural background level. Lower iron contents were found in the dust layers deposited on snow cover in



northern Utah, USA (the Wasatch Range), which varied from 1.73 to 2.85% by mass (Reynolds et al., 2014). Given the scarce information of Fe abundance available in the glacier area, we also briefly summarized the data in mineral dust from various desert regions worldwide for comparison (Table 2). The determined Fe contents in desert aerosols from ZBT (ZhengBeiTai) and Yulin in North China were

5.38% and 7.7% in total dust aerosol mass, respectively, somewhat higher than our values of cryoconite over glaciers. The reported values of Fe content from Sahara and Arabian Peninsula generally varied in the range from 2.0 to 11% by mass (Gao et al., 2001; Gomes and Gillette, 1993; Zhang et al., 2015), depending on the locations and the transport process. In addition, in this study there was no systematic variation of the Fe concentrations with altitude, which indicated that the

cryoconite on each glacier was homogeneous mixture.

Because only iron oxides (free iron) could effectively control the absorbing property of mineral materials, the content of free iron is more concerned in radiative and climate modelling. Table 2 shows the means of free and structural iron in cryoconite (percentage in total mass) from the five glaciers, with their standard deviations. Interesting, the highest value of total Fe and the highest free-to-total

ratio were found in the samples from BS. The color of BS samples was darker than others visually, and they also present the highest TOC contents among the five glaciers (Fig. 3).

The ratios of free to total iron for the five glaciers ranged from 0.31 to 0.70 (Table 2). That means substantial Fe are trapped in the crystal lattice (i.e. structural Fe) and has no direct relationship with the light absorption. This finding is generally in agreement with that of Lafon et al. (2004) for desert

aerosols (Table 2). Namely, only about half total-iron is under the form of iron oxides. Therefore, our result clearly demonstrates that the total iron is not suitable to be directly used in the albedo and radiative modelling, although this has been a common practice in previous research (Kaspari et al., 2014; Wang et al., 2013). If this point was considered, the contribution of iron-containing minerals to the total light absorption on the glacier surface will decrease almost 50%, namely, the other light-

absorbing components like black carbon and brown carbon should account for much larger fraction correspondingly.

### 3.3 The speciation of iron oxides

In previous modelling studies of the dust radiative forcing, hematite was usually assumed to be the major absorbing iron oxides (Sokolik and Toon, 1999). However, in this study goethite was found

more abundant than hematite for all the five glaciers (Fig. 4 and Table 2). The goethite in total iron oxide mass ranged from 81% (for XDK) to 98% (for BS), showing that goethite is the predominant form of iron oxides. The ratios of goethite to hematite in our study were even higher than those reported for desert aerosols (Table 2) (Formenti et al., 2014; Lafon et al., 2006; Shen et al., 2006). For example, Lafon et al (2004) reported the iron oxides in the dust from Northwest China with about half

of total iron in the form of iron oxides, and the abundance of goethite (73% of the total iron oxide mass) was higher than hematite (27%). Shen et al. (2006) determined comparable goethite/hematite composition data in dust aerosols from North China. i.e. 64% for goethite in total iron oxide mass and 36% for hematite in Dunhuang, 63% and 37% in Yulin, and 68% and 32% in Tongliao, respectively. While in the dust samples collected on the snow from American West (Wasatch Range, Utah), the



amounts of goethite and hematite in dust samples were found roughly equal using Mössbauer spectroscopy (Reynolds et al., 2014).

Beside the pedogenic characteristics, the dominance of goethite over hematite may be also ascribed to the glacier surface environment. Goethite formation is favored in moist and cool conditions, while hematite commonly occurs in warm and dry environment (Reynolds et al., 2014). That is also why the ratios of goethite to hematite were frequently used as indicators of paleoclimate (e.g. precipitation and temperature) (Schwertmann, 1971). Taking into account the cold and humid conditions on the glacier surface, mineralogical transformation of hematite to goethite is highly expected to happen, resulting more goethite.

### 3.4 Contribution to light absorption by cryoconite components

Optical properties of goethite and hematite, including MAC and AAE parameters, are critical to assess their role in the light absorption. A wide range of MAC values have been reported in the literatures. According to the previous work by Alfaro (2004), the mass absorption coefficient of iron oxides (goethite : hematite, 73% : 27%) in dust from Northwest China desert was measured as 0.56 $m^2$ $g^{-1}$ at 660 nm, and it will increase about 6 times at shorter wavelength (325 nm) (AAE ≈ 3). This value was further employed to evaluate the albedo and radiative forcing effect of dust in snow of Himalayas (Nepal) (Kaspari et al., 2014). Recently, Utry et al. (2015) reported the MAC value of hematite (purity > 95%) as 0.54 $m^2$ $g^{-1}$ at 532 nm, based on the measurements of a multi-wavelength photoacoustic instrument. Wang et al. (2013) choose the MAC of goethite of 0.9 $m^2$ $g^{-1}$ (550 nm) and AAE value of 3 to assess the contribution of mineral dust to the total absorption of LAIs in North China snow. Based on the laboratory experiments by ISSW, we determined Fe-specific absorption coefficient using the goethite (Stream Chemicals, Inc.) and hematite standard (Sigma Aldrich, Inc.) (Fig. 5). The calculated MAC values at 450 nm for goethite and hematite were 1.55 ± 0.08 $m^2$ $g^{-1}$ and 1.12 ± 0.11 $m^2$ $g^{-1}$, respectively. And the MAC values at 600 nm were 0.15 ± 0.01 $m^2$ $g^{-1}$ and 0.55 ± 0.03 $m^2$ $g^{-1}$, respectively.

Here we assumed that the total light absorption was entirely and exclusively caused by three components, i.e. iron oxides (goethite and hematite), BC and organic matters. Compared to dust and black carbon, the composition and sources of organic matters over the glacier surface are complicated. Organic matter was a mixture of soil humic and humic-like matters, biogenic particles (e.g., algae, fungi and plant debris) and biomass/fossil fuel burning emissions (Wu et al., 2016), which are often termed as brown carbon (BrC). Considering its diverse sources and complex composition, in this work we did not assume the specific optical parameters for organic matters. Instead, the relative contributions to absorption by organic matters were obtained by subtracting the portions by iron oxides and black carbon from the total absorption. The optical properties of the latter two components are much certain than light absorbing organics. The mass absorption efficiency and AAE value of BC were assumed to be 6.3 $m^2$ $g^{-1}$ (550 nm) and 1.1, respectively (Grenfell et al., 2011).

Because the light absorption capability of iron oxides and organic matters vary with wavelength, here we calculated the relative absorption of these three components at 450 nm and 600 nm, respectively. As shown in Figure 6, at 600 nm (the right panel), the organic matters dominated the light



absorption. And BC was the second contributor to the light absorption, especially for the BS glacier with the highest BC concentration. However, the contribution of iron oxides increases dramatically at 450 nm, especially for goethite, due to their high light absorption ability at short wavelength. For the glaciers except BS, the absorption by goethite was larger than BC and approximately equal to organic matters. The increased contribution by goethite at the shorter wavelength was due to its large AAE value (Zhou et al., 2017), which indicated stronger light absorption at short wavelength. While the relative contribution to absorption by hematite appeared to be constant between different wavelengths.

In general, goethite plays a stronger role at shorter band, causing higher fraction of light absorption than BC. Although iron oxides are much less absorbing than black carbon per unit mass, the much high mass concentration of mineral dust in the natural environment may result in total absorption be larger than BC.

## 4. Summary and Conclusions

The degree to which mineral dust, especially iron oxides, affect the solar absorption and decreases of snow albedo remain uncertain. Despite their importance, the content and speciation of iron oxides in the cryoconite over the glacial surface has not been reported previously.

The iron abundances in the cryoconite from TP glaciers ranged from 3.40% to 4.90% by mass, which were comparable to the upper continental crust (UCC) composition (3.5% of Fe) (Taylor and McLennan, 1995) and implied their natural sources. We further separated and determined iron oxides (free Fe) using the Citrate-Bicarbonate-Dithionite method. The ratios of free to total iron for the five glaciers ranged from 0.31 to 0.70. That means substantial amounts of Fe are trapped in the crystal lattice (i.e. structural Fe) and has no direct influence on the light absorption. Our result clearly demonstrated that the total iron was not suitable to be directly used in the albedo and radiative modelling, although this is a common practice in previous studies. The iron oxides were further quantified into goethite and hematite, the two major species. The goethite content in iron oxides (in mass fraction) ranged from 81% (XDK) to 98% (BS), showing that goethite is the predominant form of iron oxides.

Taking account of both the abundance of iron oxides and their optical properties, the total light absorption were quantitatively attributed to goethite, hematite, BC and organic matters at 450 nm and 600 nm. Organic matters were found to be the most important light absorber at 450 nm and 650 nm wavelengths. We demonstrated that the goethite played a stronger role than BC at shorter wavelength (i.e. 450 nm) for glaciers except Baishui (BS) glacier. Baishui glacier is closely adjacent to the intensive human activities area, and receives more BC from the emission there. In general, this research provided new observations of the iron-oxides in glaciers, and the results are meaningful for understanding their role on mountain glacier surfaces in Himalayas and Tibetan Plateau, a climate sensitive and environmentally fragile region.



**Acknowledgements**

We deeply thank Wei YANG, Yajun LIU, Hewen NIU, Junming GUO, Zhiwen DONG, Xiaofei LI, Yang LI and other team members for their contribution in the field sampling. This work is supported by National Science Foundation of China under Grants 41522103, 41522505, 41225002 and 41673095.

The data used are listed in the references, tables, and supplements.

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



**Table 1 Summary of cryoconite samples collected from five glaciers over Tibetan Plateau and surroundings.**

| Glacier | Description | Coordinates | Elevation (m, a.s.l.) | Sampling date | Sample Number |
|---|---|---|---|---|---|
| **Urumqi #1 Glacier (UG)** | Tienshan Mount. | 43°06′N, 86°48′E | 3800-4000 | Aug. 2014 | 14 |
| **LaoHuGou (LHG)** | Qilian Mount. | 39°28′N, 96°32′E | 4300-4900 | Jul. 2014 | 14 |
| **Xiaodongkemadi (XDK)** | Tanggula Mount. | 33°03′N, 92°04′E | 5400-5600 | Aug. 2014 | 6 |
| **PaLong #4 (PL)** | Southeast TP | 29°15′N, 96°56′E | 4700 | Sep. 2015 | 2 |
| **Baishui #1 Glacier (BS)** | YuLong | 27°6′N , 100°11′E | 4600-4800 | Aug. 2014 | 9 |





**Table 2 The abundances (by mass) of total iron, free iron, hematite, goethite and their ratios determined in cryoconites from TP glaciers, as well as other data available in the literatures.**

| Locations | Description | Total iron (%) | Free iron (%) | Free-total iron ratio | Hematite % | Goethite % | Gt-Hm ratio | References |
|---|---|---|---|---|---|---|---|---|
| UG | | 4.62 (±0.22)[a] | 1.41 (±0.29) | 0.31 (±0.07) | 0.24 (±0.02) | 1.98 (±0.47) | 8.26 (±2.27) | This study |
| LHG | | 4.28 (±0.17) | 1.50 (±0.30) | 0.35 (±0.07) | 0.28 (±0.02) | 2.08 (±0.48) | 7.54 (±1.83) | This study |
| XDK | Cryoconite | 3.40 (±0.18) | 1.93 (±0.48) | 0.56 (±0.12) | 0.56 (±0.08) | 2.44 (±0.69) | 4.30 (± 0.90) | This study |
| PL | | 4.18(±0.13) | 1.53(±0.28) | 0.37(±0.08) | 0.23(±0.01) | 2.18(±0.44) | 9.63(±1.88) | This study |
| BS | | 4.90 (±0.21) | 3.43 (±0.53) | 0.70 (±0.10) | 0.10 (±0.01) | 5.35 (±0.85) | 55.2 (±10.4) | This study |
| Utah, USA | Dust on snow | 1.73-2.85 | | | | | ~1[b] | Reynolds et al. (2014) |
| Niger, Sahara | Desert Aerosol | 6.3 (±0.9) | 2.8 (±0.8) | 0.44 (±0.11) | | | | Lafon et al. (2004) |
| Niger, Sahel | Desert Aerosol | 7.8 (±0.4) | 5.0 (±0.4) | 0.65 (±0.04) | | | | Lafon et al. (2004) |
| Yulin, China | Desert Aerosol | 7.7 (±0.3) | 3.7 (±0.4) | 0.48 (±0.03) | | | | Lafon et al. (2004) |
| ZBT, China | Desert aerosol | 5.38 (±0.2) | 3.0 (±0.2) | 0.43 (±0.01) | | | ~3.0 | Lafon et al. (2006) |
| West Africa | Desert Aerosol | | | 0.38-0.72 | 0.09-0.26[c] | 0.21-0.49[c] | 0.96-3.1 | Formenti et al. (2014) |

[a] Values in brackets represent the standard deviations; [b] Mössbauer spectroscopy indicated roughly equal amounts of hematite and goethite, while reflectance spectroscopy showed goethite was dominant iron oxides; [c] Using X-ray absorption (XAS). Note: the total and free iron data in this table were obtained from ICP-MS, so it refers to the elemental Fe. While hematite and goethite were measured as Fe oxides.

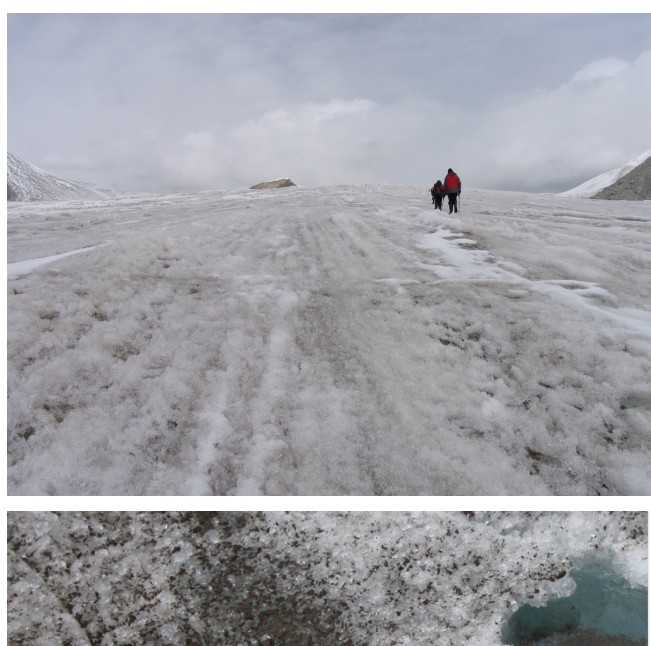

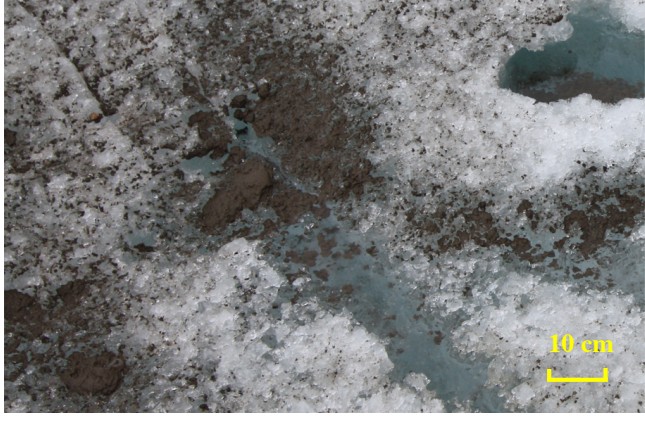

**Fig. 1 Surface of mountain glacier (upper) in central Tibetan Plateau and dispersed cryoconite (down) on it.**



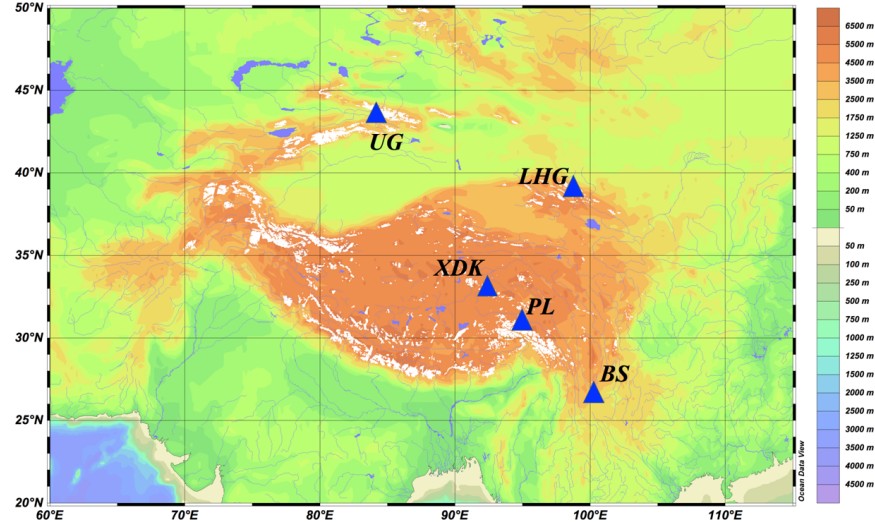

**Fig. 2 Topographic map of the Tibetan Plateau and surrounding, with locations of five representative glaciers. Note: the base map was created by Ocean Data View software (Schlitzer, 2017).**



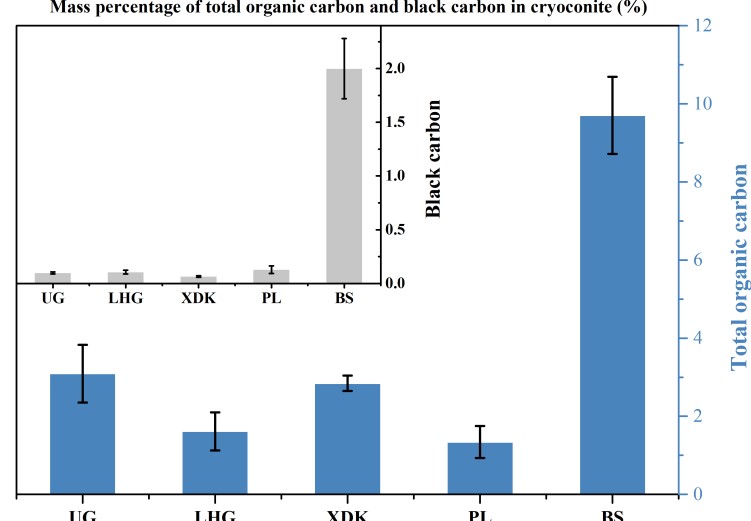

**Fig. 3** The total organic carbon (blue) and black carbon (grey) in the total mass of cryoconites on the mountain
glaciers of Tibetan Plateau and surroundings.



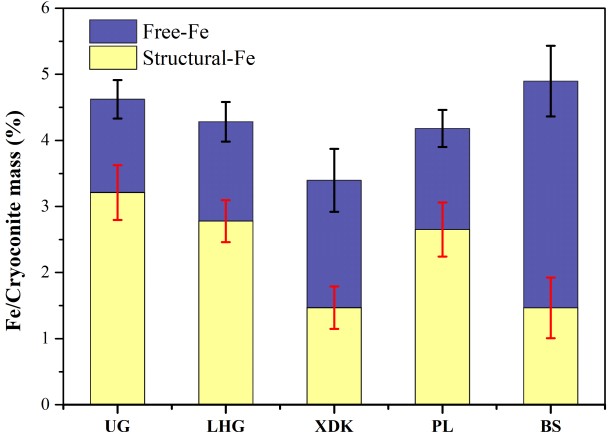

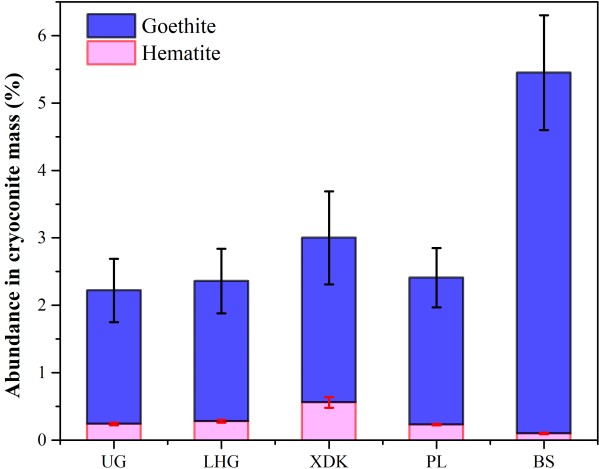

**Fig. 4 The free iron and structural iron contents (elemental Fe) measured in the total cryoconite (dried) mass from TP glaciers (Upper), and goethite and hematite (Fe oxides) contents in the total cryoconite mass (Bottom).**





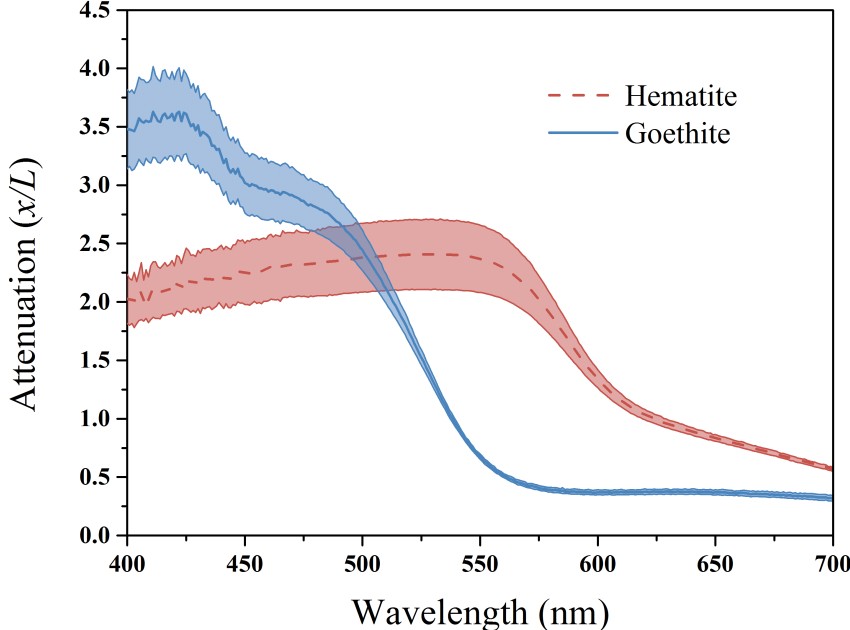

**Fig. 5 The mass weighted light attenuation by hematite and goethite. Error bars indicate the standard deviation.**





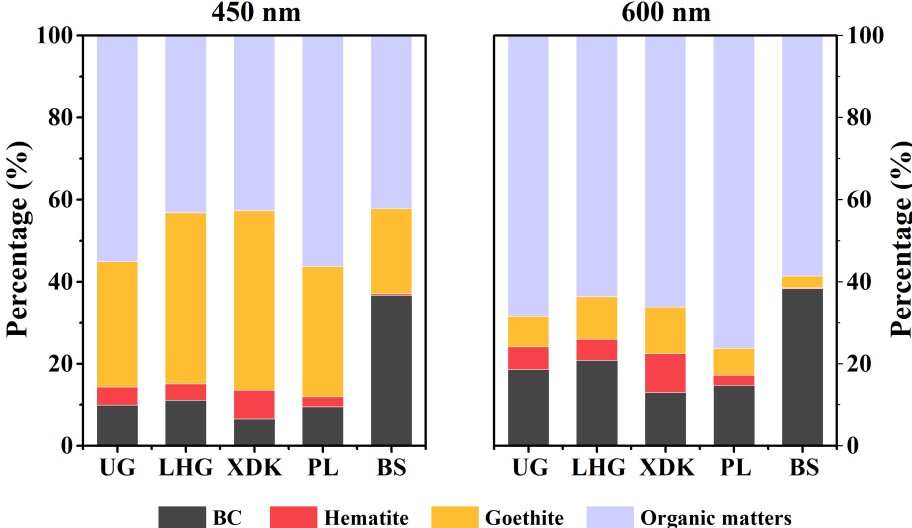

Fig. 6 Apportioning of total light absorption (450 nm and 600 nm, respectively) to black carbon, hematite, goethite

5 and organic matters for the cryoconite from five TP glaciers.