# Peer review of "Iron oxides in the cryoconite on the glaciers over Tibetan Plateau: abundance, speciation and implications"

_The Cryosphere, 2018_

## Referee Comment (RC1) · Anonymous Referee #1 · 4 Jun 2018

This paper concerns iron geochemistry in cryoconite samples from the Tibetan plateau region. Its main focus is on the optical impacts related to iron oxides on the properties of cryoconite and of its potential role in the reduction of glacial albedo, also considering other impurities that could play a role in this context: organic and inorganic carbon and dust. The topic is definitely appropriate for The Cryosphere. Unfortunately, I have some concerns about the methodological side of this work. This is a paper where the experimental side is dominant, since many measurements were carried out, using different instruments and techniques. For this reason I would have expected that the discussion about accuracy, precision, reproducibility was expanded and largely detailed. On the contrary it is poor and the reader cannot understand and evaluate the significance and

the robustness of the data.

One of the most critical points concerns sample preparation for ICP-MS. The authors declare that the samples were prepared for ICP-MS analyses using 1% HNO3. This is impossible. Dealing with mineral samples it is necessary to completely dissolve them using high concentration inorganic acids. Nitric acid alone is not sufficient and surely if used at the concentration of 1%. For example, there are a lot of mineral phases that can be dissolved only using a mixture of HClO4, HF and HNO3. If the authors applied the protocol they described, results cannot be considered reliable and I also cannot believe that their recovery factor for Fe was 95 %. Using 1% HNO3 is almost the same of using pure water. The authors should explain in detail this point. In addition to this, it should be taken into account that the authors applied a stronger acid attack to the samples for carbon analysis, I am asking myself why they didn't apply the same protocol for ICP-MS measurements.

Still on the methodologies. I have some concerns about their method to estimate the mineral composition of Fe oxides. At first it should be mentioned that if the datum about total iron content (i.e. the one gathered through ICP-MS) is wrong, all the successive analysis about "free iron-total iron-oxide iron" is in turn inaccurate. Secondarily the equation presented at page 4, line 26, is given without any explanation or reference and it is the equation that allowed the authors distinguishing goethite from hematite. What is strange is that the authors used only hematite to test and validate their method, but they are working on both the oxides. Given the fact that one of the main result of this work is that Fe oxide contained in cryoconite is almost completely composed by goethite (more than 80%), it is strange that they prepared their calibration using only hematite, which accounts only for less than 20 % of their samples.

Given these critical issues I cannot support the publication of this paper in The Cryosphere. The authors should completely revise their methodological approach before going on with the analysis and the interpretation of the data.

Despite my final comment, I started to comment more punctually the paper. For this reason, I include a partial revision of the paper.

PAGE 1

Line 18: change to "influence the radiative properties of mineral dust and thus its radiative impact. In particular, the different optical features of. . ."

Line 20: the term speciation is not appropriate here, you are talking about minerals, not elements. You could refer to "geochemical behavior" or something like that; change to "from five glaciers located in different regions of the Tibetan. . ."

Line 21: "abundance"

Line 22: change to ". . . by mass, in accordance to typical natural background level"

Line 23-25: the passage is not clear, why finding such ratios should be indicative about considering free or immobile iron fractions? Probably something is missing here.

Line 25-27: I guess that here you are referring to the only immobile fraction of Fe, aren't you? So probably it would be better to change to "Considering the immobile mineral Fe fraction, goethite is definitely dominant, accounting for more than 80 % of total iron".

Line 31: change to "anthropogenic/natural impacts on glaciers."; remove "and then taking the proper mitigation measures."

Line 33: they can do that, not could

Line 34: you talk about glacier and snowpack, what about ice?; change to "more solar energy, with effects on glacier mass balance (Warren and Wiscombe, 1985)

Line 35: improve references, there are plenty of good works about this point, not only one; remove "along with the rising air temperatures (IPCC, 2014)"

Line 37: what is soil dust? On the surface of glaciers it is more common to find rock fragments or dust produced from the weathering of the surrounding rocky outcrops. In

addition to this you should also considered long-range transport from arid areas. See Cook et al., 2016 to review this point ("The dark biological secret of the cryosphere").

PAGE 2

Line 1-4: I don't understand why the compositional complexity of TP cryoconite should be major than the one of cryoconite from other areas. Be careful because Baccolo et al., 2017 talks about Alpine cryoconite, not about the TP.

Line 5: change to: "Considering this region, tremendous attention has been paid to..."

Line 6: change "partly" with "mainly"

Line 7: remove "and is receiving strong influences from those anthropogenic Emissions"

Line 8: change "Nevertheless" with "On the contrary"; change to "despite dust is apparently the predominant..."

Line 12: change to "the snow albedo reduction and the subsequent radiative forcing caused by dust overwhelm the impact related to black carbon.

Line 14: "models"

Line 15: change to: "only dust concentration is taken into account, not its composition"

Line 16-22: consider also Formenti et al., 2014 ("Dominance of goethite over hematite in iron oxides of mineral dust from Western Africa: Quantitative partitioning by X‐ray absorption spectroscopy") and references therein. You are saying that these two oxides are the most common ones "in nature", but this true only if you talk about atmospheric mineral dust, not if you consider the entire Earth (see for example Torrent et al., 1983 "Quantitative relationships betwe en soil color an hematite content"). Rewrite this passage.

Line 23-24: Why you say "are not well understood"? You could say that they are not

investigated, not that they are not understood. Do you know a recent paper from Hawkings et al., 2018 ? ("Biolabile ferrous iron bearing nanoparticles in glacial sediments"), I guess you could find useful information in this sense.

Line 25: "remains"; change to "The aim of the present work is to estimate the radiative forcing of cryoconite in the TP region, trying to address several key issues."

Line 34: change "examine" with "consider"; "regions"; remove "and surroundings"

Line 35: change to "were chosen to sample the cryoconite"; change to "A detailed description of the collection sites is given in Table 1"

Line 36: change to "The Urumqi No.1 glacier (UG, 43°06′N, 86°49′E), presents two branches covering 1.6 km2, it is located in eastern Tien Shan. The air circulation of this region is dominated by westerly winds in summer and by the influence of the Siberian High baric field during winter (Wang et al, 2014)."

PAGE 3

Line 2: change to "(LHG, XX°XX'X, XX°XX'X)

Line 3: change to "where a typical continental climate is found (Dong et al., 2014)"

Line 5: follow the sam scheme of above "(XDK, XX°XX'X, XX°XX'X)"

Line 6: "at the center of TP"

Line 7: "Tanggula Mountains represent the northern boundary of the area influenced by the South Asian monsoon."

Line 8: "(PL, XX°XX'X, XX°XX'X)"

Line 10: "and it is characterized"

Line 11: "(BS, XX°XX'X, XX°XX'X)"; "with a length of 2.26 km and an area of 1.32 km2"

Line 12: "in the Yulong Mountains, at the southeastern edge of TP"

Line13: "low altitude"

Line 14: "kept frozen"

Line 17: "freeze-dried" what dou you mean, explain; "and reduced to powder"; how did you powdered your samples? Explain also this point

Line 18: "Therefore in this work concentrations and fractions are referred to dry cryoconite mass."

Line 21: change to "under a laboratory bench"

---

## Author Comment (AC1) · 1 Jul 2018

We would like to thank the reviewer for the time and effort reviewing the manuscript. The comments are very thoughtful and helpful for improvement of our manuscript. We have listed all reviewer comments below and our answers are provided in blue. A version of the revised manuscript was provided as a supplement.

**Anonymous Referee #1**

This paper concerns iron geochemistry in cryoconite samples from the Tibetan plateau region. Its main focus is on the optical impacts related to iron oxides on the properties of cryoconite and of its potential role in the reduction of glacial albedo, also considering other impurities that could play a role in this context: organic and inorganic carbon and dust. The topic is definitely appropriate for The Cryosphere. Unfortunately, I have some concerns about the methodological side of this work. This is a paper where the experimental side is dominant, since many measurements were carried out, using different instruments and techniques. For this reason I would have expected that the discussion about accuracy, precision, reproducibility was expanded and largely detailed. On the contrary it is poor and the reader cannot understand and evaluate the significance and the robustness of the data.

**Reply:** Thank you for the comments. We totally agree with you that the data quality is the essential of the research. Now we tried our best to provide the quality control information for each measurement, involving accuracy, precision and reproducibility. Please check the details in the revised manuscript.

One of the most critical points concerns sample preparation for ICP-MS. The authors declare that the samples were prepared for ICP-MS analyses using 1% HNO3. This is impossible. Dealing with mineral samples it is necessary to completely dissolve them using high concentration inorganic acids. Nitric acid alone is not sufficient and surely if used at the concentration of 1%. For

example, there are a lot of mineral phases that can be dissolved only using a mixture of HClO4, HF and HNO3. If the authors applied the protocol they described, results cannot be considered reliable and I also cannot believe that their recovery factor for Fe was 95 %. Using 1% HNO3 is almost the same of using pure water. The authors should explain in detail this point. In addition to this, it should be taken into account that the authors applied a stronger acid attack to the samples for carbon analysis, I am asking myself why they didn't apply the same protocol for ICP-MS measurements.

**Reply:** Thank you very much for pointing out this critical problem. Due to our carelessness, this sentence appears misleading. Actually, here 1% $HNO_3$ means the samples were conditioned in 1% $HNO_3$ solution finally after digestion, and ready for the subsequent ICP-MS analysis. For the digestion, we did use strong acids to treat the cryoconite samples. Now we added the detailed description of the sample pretreatment as following (Page 3, Line 23-38 in the revised manuscript):

*A portion of cryoconite sample (about 20 mg) was dissolved under a laboratory hood using HF+HNO₃ mixture through three steps. In details, the sample was firstly transferred into PTFE high-pressure digestion vessel, and 1 mL HF and 1 ml HNO₃ were added. The digestion vessel was then ultrasonic treated for 20 min, and evaporated to nearly dry on a hot plate. Another 1 mL HF and 1 ml HNO₃ were added, and digested in an oven at 190 °C for 24 h. After cooling, the vessel was opened and evaporated to nearly dry again (on the plate at 150 °C), then followed by a second addition of HNO₃. This procedure was repeated to wipe off the HF completely. Then 1mL HNO₃ and 3mL H₂O were added to the vessel and put it into the oven for another 24h at 150 °C. After cooling, the final solution was diluted with pure water to about 50mL.*

*Eventually, it was measured by inductively coupled plasma-mass spectrometry (ICP-MS, Thermo X7, Thermo-Elemental Corp.) for Fe and other elements. Indium, Rhodium and Rhenium solution were used as internal*

*standards. The accuracy and precision of trace elements was ascertained based on repeated measurement of the USGS Geochemical Reference Standard (Andesite, AGV-2). The measured and certified values for Fe agree well, with recovery better than 95%. The detailed description of the analytical protocol in our laboratory could be found in previous work (Wu et al., 2009).*

Still on the methodologies. I have some concerns about their method to estimate the mineral composition of Fe oxides. At first it should be mentioned that if the datum about total iron content (i.e. the one gathered through ICP-MS) is wrong, all the successive analysis about "free iron-total iron-oxide iron" is in turn inaccurate. Secondarily the equation presented at page 4, line 26, is given without any explanation or reference and it is the equation that allowed the authors distinguishing goethite from hematite. What is strange is that the authors used only hematite to test and validate their method, but they are working on both the oxides. Given the fact that one of the main result of this work is that Fe oxide contained in cryoconite is almost completely composed by goethite (more than 80%), it is strange that they prepared their calibration using only hematite, which accounts only for less than 20 % of their samples.

**Reply:** Just as explained above, we believed that the total iron content determined by ICP-MS in this study is reliable. Furthermore, hematite and goethite are the two main coloring agents in mineral dusts and are characterized by distinct colors, red and yellow, respectively. Hematite was chosen to be quantified rather than goethite, because hematite is a more intense (effective) coloring agent than goethite. The limit of detection for Hm can be as low as 0.01% by weight (Balsam et al., 2014; Deaton and Balsam, 1991; Ji et al., 2002; Lu et al., 2017). Now we added this explanation in the main text (please see the changes in Page 4, Line 30-34) with the relevant references.

PAGE 1  Line 18: change to "influence the radiative properties of mineral dust and thus its radiative impact. In particular, the different optical features of. . ."
**Reply:** Changed.

Line 20: the term speciation is not appropriate here, you are talking about minerals, not elements. You could refer to "geochemical behavior" or something like that; change to "from five glaciers located in different regions of the Tibetan. . ."
**Reply:**   Changed.

Line 21: "abundance" Line 22: change to ". . . by mass, in accordance to typical natural background level"
**Reply:**   Changed.

Line 23-25: the passage is not clear, why finding such ratios should be indicative about considering free or immobile iron fractions? Probably something is missing here.
**Reply:**   Changed.

Line 25-27: I guess that here you are referring to the only immobile fraction of Fe, aren't you? So probably it would be better to change to "Considering the immobile mineral Fe fraction, goethite is definitely dominant, accounting for more than 80 % of total iron".
**Reply:**   Here, it means goethite content (%) in iron oxides (i. e. free iron fraction).

Line 31: change to "anthropogenic/natural impacts on glaciers."; remove "and then taking the proper mitigation measures."
**Reply:**   Changed.

Line 33: they can do that, not could
**Reply:**   Changed.

Line 34: you talk about glacier and snowpack, what about ice?; change to "more solar energy, with effects on glacier mass balance (Warren and Wiscombe, 1985)
**Reply:** Usually, the glacier surface is covered by snowpack. So, we used snowpack instead of bare ice.

Line 35: improve references, there are plenty of good works about this point, not only one; remove "along with the rising air temperatures (IPCC, 2014)"

**Reply:** The reference cited here is a comprehensive review paper, which was contributed by different groups in this field, and well reflected the recent progress on this issue. Now we changed it to (Qian, et al., 2015 and the references therein).

Line 37: what is soil dust? On the surface of glaciers it is more common to find rock fragments or dust produced from the weathering of the surrounding rocky outcrops. In addition to this you should also considered long-range transport from arid areas. See Cook et al., 2016 to review this point ("The dark biological secret of the cryosphere").

**Reply:** Yes, the dust over glacier surface could originate from different sources, not only the soil. To make it clear, now we use "mineral dust". In this work, our focus is not to differentiate the long-range transport or local source of dust.

PAGE 2

Line 1-4: I don't understand why the compositional complexity of TP cryoconite should be major than the one of cryoconite from other areas. Be careful because Baccolo et al., 2017 talks about Alpine cryoconite, not about the TP.

**Reply:** Here we cited the reference of Baccolo et al., 2017, with the aim to give a definition of cryoconite. For the mountainous glaciers, the cryoconite at Alpine and Tibetan Plateau share similarities, although specific components may vary.

Line 5: change to: "Considering this region, tremendous attention has been paid to. . ." Line 6: change "partly" with "mainly"

**Reply:** Changed.

Line 7: remove "and is receiving strong influences from those anthropogenic Emissions"

**Reply:** Changed.

Line 8: change "Nevertheless" with "On the contrary"; change to "despite dust is apparently the predominant. . ."
**Reply:** Changed.

Line 12: change to "the snow albedo reduction and the subsequent radiative forcing caused by dust overwhelm the impact related to black carbon.
**Reply:** Changed.

Line 14: "models"
**Reply:** Changed.

Line 15: change to: "only dust concentration is taken into account, not its composition"
**Reply:** We intend to keep the current expression.

Line 16-22: consider also Formenti et al., 2014 ("Dominance of goethite over hematite in iron oxides of mineral dust from Western Africa: Quantitative partitioning by X-ray absorption spectroscopy") and references therein. You are saying that these two oxides are the most common ones "in nature", but this true only if you talk about atmospheric mineral dust, not if you consider the entire Earth (see for example Torrent et al., 1983 "Quantitative relationships between soil color an hematite content"). Rewrite this passage.
**Reply:** The reference (Formenti et al., 2014) has been added here. And we changed "in nature" into "mineral dust".

Line 23-24: Why you say "are not well understood"? You could say that they are not investigated, not that they are not understood. Do you know a recent paper from Hawkings et al., 2018? ("Biolabile ferrous iron bearing nanoparticles in glacial sediments"), I guess you could find useful information in this sense.
**Reply:** Thank you for this point. The recent work by Hawkings and colleagues studied the iron speciation and bioavailability (Fe(II) and Fe(III)-

bearing nanoparticles) in Arctic glacial area. Now this paper (Hawkings et al., 2018) was cited to better reflect the progress on this subject.

Line 25: "remains"; change to "The aim of the present work is to estimate the radiative forcing of cryoconite in the TP region, trying to address several key issues."

**Reply:** Changed. Actually, the present work did not estimate the radiative forcing directly, while only constrained the uncertainties through the iron study.

Line 34: change "examine" with "consider"; "regions"; remove "and surroundings"

**Reply:** Changed. But we intend to keep "and surroundings", because one site in this study (Urumqi No. 1 Glacier) does not belong to Tibetan Plateau. So "surroundings" is more appropriate regarding the geographic definition.

Line 35: change to "were chosen to sample the cryoconite"; change to "A detailed description of the collection sites is given in Table 1"

**Reply:** Changed.

Line 36: change to" The Urumqi No.1 glacier (UG, 43°06′N, 86°49′E), presents two branches covering 1.6 km2, it is located in eastern Tien Shan. The air circulation of this region is dominated by westerly winds in summer and by the influence of the Siberian High baric field during winter (Wang et al, 2014)."

**Reply:** We intend to keep the original expression.

PAGE 3

Line 2: change to "(LHG, XX°XX'X, XX°XX'X)

**Reply:** Changed.

Line 3: change to "where a typical continental climate is found (Dong et al., 2014)"

**Reply:** We intend to keep the original expression.

Line 5: follow the same scheme of above "(XDK, XX°XX'X, XX°XX'X)"

**Reply:**    Changed.

Line 6: "at the center of TP"

**Reply:**    Changed.

Line 7: "Tanggula Mountains represent the northern boundary of the area influenced by the South Asian monsoon."

**Reply:**    Changed.

Line 8: "(PL, XX°XX'X, XX°XX'X)" Line 10: "and it is characterized" Line 11: "(BS, XX°XX'X, XX°XX'X)"; "with a length of 2.26 km and an area of 1.32 km2" Line 12: "in the Yulong Mountains, at the southeastern edge of TP"

**Reply:**    Changed.

Line13: "low altitude" Line 14: "kept frozen"

**Reply:**    We intend to keep the original expression.

Line 17: "freeze-dried" what do you mean, explain; "and reduced to powder"; how did you powdered your samples? Explain also this point

**Reply:**    Freeze-dried is a low temperature dehydration process, which involves freezing the samples and then removing the water (ice) by sublimation. It is commonly used in the laboratory. Here, the cryoconite samples turn into powder naturally after being freeze-dried.

Line 18: "Therefore in this work concentrations and fractions are referred to dry cryoconite mass."

**Reply:**    Changed.

Line 21: change to "under a laboratory bench"

**Reply:**    Changed.

**Thank you again for your valuable comments and suggestions.

**Reference**

[revised manuscript text omitted]

---

## Referee Comment (RC2) · Anonymous Referee #2 · 2 Jul 2018

The paper investigates the abundance, speciation and spectral absorption of iron oxides in five glaciers in the Tibetan Plateau. Samples were collected on the field and analyzed in the laboratory to retrieve their composition in terms of iron oxides, black carbon and organic matter. Measurements of the spectral absorption were performed on the collected samples and the partitioning of the absorption due to mineral dust, black carbon, and organic material was estimated. The study is quite an interesting contribution and in my opinion it deserves publication in "The Cryosphere". I have nonetheless few comments concerning the data treatment and discussion that I will detail in the following. I have in particular some doubt on the choices performed to treat absorption measurements and I would like the authors to improve this part by

adding more details and by performing some sensitivity calculations. Also the discussion would benefit of some more specific comments on the representativeness and implications of the results. This is why I suggest major revisions for the paper. A significant discussion on the part concerning chemical analysis has been already performed with regard to the comments of the other reviewer and I will not add other comments on this part.

Comments:

Abstract: please define what Cryoconite is also in the abstract

Section 2.6: In the procedure for MAC estimate of cryoconite you need to make assumptions on the MAC and you assume the MAC of fullerene, i.e. a proxy for BC, in your calculations. First, I am not sure to completely understand the procedure followed to retrieve the MAC of cryoconite and I would ask the authors to provide more details on this part; second, I wonder: which is the impact of the assumption on the MAC on the obtained results? I mean, what is the uncertainty in the retrieved MAC of cryoconite due to the fact of assuming the MAC of fullerene in calculations? It would have not been more appropriate to use a weighted average MAC between BC, dust, and organics based on their mass contribution to cryoconite estimated deposits? For a reference of the MAC of dust see for example the recent paper by Caponi et al. (2017).

Caponi, L., Formenti, P., Massabó, D., Di Biagio, C., Cazaunau, M., Pangui, E., Chevaillier, S., Landrot, G., Andreae, M. O., Kandler, K., Piketh, S., Saeed, T., Seibert, D., Williams, E., Balkanski, Y., Prati, P., and Doussin, J.-F.: Spectral- and size-resolved mass absorption efficiency of mineral dust aerosols in the shortwave spectrum: a simulation chamber study, Atmos. Chem. Phys., 17, 7175-7191, https://doi.org/10.5194/acp-17-7175-2017, 2017.

Always concerning Sect. 2.6, if available, it would have not been useful also to calibrate light attenuation against pure hematite and goethite minerals? This point has been probably already raised by the other reviewer, but I repeat the question.

Sections 3.3 and 3.4: take into account the Caponi et al. (2017) reference values for the MAC in the calculations. Also iron oxide and their speciation for dust samples from many regions worldwide was reported in that work, and these data can be useful for your data interpretation.

Sections 3.3 and Conclusions: I guess one interesting point to discuss based on your results and the comparison with the literature is the regional scale variability of iron content and its speciation and the impact on glacier absorptivity and albedo. I would develop this aspect more in the discussion. Could you also add some calculations of how much spectral albedo would change in relation to absorption by different species as found in your study? What about the seasonal and spatial representativeness of your data?

Section 3.3, Page 7, line 8: do you mean the effect of atmospheric aging on minerals? Please be more specific.

Page 8, lines 8-11: I guess this is basically your key conclusion and I would move it to Sect. 4. Also I suggest to add a brief discussion on the impact of the regional variability of iron oxides and their speciation, and the representativeness of your results compared to other regions of the world under the influence of other deserts with different mineralogical compositions.

---

## Author Comment (AC2) · 20 Jul 2018

We really thank the reviewer for the positive and helpful comments. Now the manuscript has been revised accordingly. Our responses are written in blue text. A version of the revised manuscript was provided as a supplement.

**Anonymous Referee #2**

1. The paper investigates the abundance, speciation and spectral absorption of iron oxides in five glaciers in the Tibetan Plateau. Samples were collected on the field and analyzed in the laboratory to retrieve their composition in terms of iron oxides, black carbon and organic matter. Measurements of the spectral absorption were performed on the collected samples and the partitioning of the absorption due to mineral dust, black carbon, and organic material was estimated. The study is quite an interesting contribution and in my opinion it deserves publication in "The Cryosphere". I have nonetheless few comments concerning the data treatment and discussion that I will detail in the following. I have in particular some doubt on the choices performed to treat absorption measurements and I would like the authors to improve this part by adding more details and by performing some sensitivity calculations.

**Reply**: Thank you for the recognition of our work. We have added more details about the absorption measurements and performed sensitivity calculations in the manuscript. "Because the ISSW spectrophotometer is only sensitive to the signal of light absorption but not scattering of LAIs on filters due to its integrating sandwich structure, he light attenuation by samples on filter at specific wavelength will be converted to equivalent BC mass loading" (See Line 11-14, Page 5). "The standard deviation between repeated measurements range from 0.29% to 7.83% (mean value: 2.92%) in this study, which falling into the recommended variation range (i.e., < 10%) by Grenfell et al. (2011)." (See Line 23-24, Page 5).

2. Abstract: please define what Cryoconite is also in the abstract.
**Reply**: Changed. (See Line 17-18, Page 1)

3. Section 2.6: In the procedure for MAC estimate of cryoconite you need to make assumptions on the MAC and you assume the MAC of fullerene, i.e. a proxy for BC, in your calculations. First, I am not sure to completely understand the procedure followed to retrieve the MAC of cryoconite and I would ask the authors to provide more details on this part.

Reply: Thanks for this comment. The ISSW method we used is originally established by Grenfell et al. (2011) and subsequently employed to analyze the light absorbing impurities in snow from North American (Dang and Hegg, 2014), Northern China (Wang et al., 2013) and Arctic (Forsstrom et al., 2013). The ISSW spectrophotometer measures the light attenuation spectrum by LAIs on filters. Then, the light attenuation was calibrated against a set of standard BC samples (i.e., fullerene) to determine the equivalent BC loading, which could produce the same amount of light attenuation by LAIs on filter. Because the ISSW system is sensitive to light absorption but not scattering owning to the enhanced absorption signal result from its diffuse radiation field and integrating sandwich sphere (Grenfell et al., 2011), the absorption optical depth of LAIs could derived from the multiplication between equivalent BC loading and its MAC (0.63 $m^2$ $g^{-1}$ at 550 nm). The MAC of cryoconite was then calculated from dividing the optical depth by the mass loading of LAIs.

Second, I wonder: which is the impact of the assumption on the MAC on the obtained results? I mean, what is the uncertainty in the retrieved MAC of cryoconite due to the fact of assuming the MAC of fullerene in calculations? It would have not been more appropriate to use a weighted average MAC between BC, dust, and organics based on their mass contribution to cryoconite estimated deposits? For a reference of the MAC of dust see for example the recent paper by Caponi et al. (2017).

Reply: According to the previous research, the relative uncertainty in MAC of fullerene is no more than 6.6% by the ISSW method (Zhou et al., 2017). And based on the study by Grenfell et al. (2011), the main uncertainties in this method derived from instrumental uncertainty and nonuniformities in the sample deposition on filter, which could be evaluated by repeated measurement. In our study, the standard deviation

between repeated measurement range from 0.29% to 7.83% (mean value: 2.92%) (See Line 23-24, Page 5), which falls into the recommended variation range (i.e., < 10%) by Grenfell et al. (2011). This demonstrate the high reliability of our method used to analyze the optical properties of cryoconite.

Regarding the second question, as our understanding on your suggestions, the weighted average MAC of cryoconite could be calculated by the following formula:

$$MAC_{cyroconite} = MAC_{dust} \cdot f_{dust} + MAC_{BC} \cdot f_{BC} + MAC_{OM} \cdot f_{OM}$$

However, the relative mass concentration of organic matters in cyroconite ($f_{OM}$) is unknown, although the BC and iron oxides were determined. More important, the light absorption properties of organics, such as AAE, could range from 2 to 11 at different environments (Laskin et al., 2015), due to their complicated sources and composition (HULIS or algae). Therefore, it will lead to high uncertainty in the weighted average MAC of cryoconite if we simply assume a MAC value for organic matters. Considering the two reasons above, we intend to obtain the MAC of cryoconite directly using the ISSW method.

4. Always concerning Sect. 2.6, if available, it would have not been useful also to calibrate light attenuation against pure hematite and goethite minerals? This point has been probably already raised by the other reviewer, but I repeat the question.

**Reply**: Thanks for this comment. Actually, using standard black carbon (i.e. fullerene) is well established for the ISSW measurement. In details, the calibration is based on a set of seven standard filters with a series of loadings of fullerene, a commercially produced soot. These were prepared through sequential dilutions and gravimetric confirmation of a standard soot suspension obtained after previous filtration through 2.0 µm and 0.8 µm pore Nuclepore filters (Grenfell et al., 2011). The filter loadings span a range sufficient to define the instrument sensitivity curves for field samples over the full visible spectral band of interest. In contrast, these reference filters by hematite or goethite is not available currently.

5. Sections 3.3 and 3.4: take into account the Caponi et al. (2017) reference values for the MAC in the calculations. Also, iron oxide and their speciation for dust samples from many regions worldwide were reported in that work, and these data can be useful for your data interpretation.

**Reply**: Thank you for this valuable comment. We have considered the results in the recommended paper during the discussion. (See Line 16-17, Line 31-32, Page 6; Line 4-6, Page 8).

6. Sections 3.3 and Conclusions: I guess one interesting point to discuss based on your results and the comparison with the literature is the regional scale variability of iron content and its speciation and the impact on glacier absorptivity and albedo. I would develop this aspect more in the discussion. Could you also add some calculations of how much spectral albedo would change in relation to absorption by different species as found in your study? What about the seasonal and spatial representativeness of your data?

**Reply**: Thanks. We agree with the reviewer that it will be better to calculate the radiative effect of LAIs in cryoconite. Unfortunately, it is beyond our ability to do this in this work. We are focusing on the chemical compositions and light absorption parameters (i.e., MAC and AAE) of LAIs in this study. Currently, due to the limitation of our field sampling, we can not obtain the spatial distribution of cryoconite on the glacier surface (i.e. the percentage of cryoconite cover in a unit of glacier surface area). In the future research, we will explore quadrat sampling to reveal the exact change of albedo by cryoconite. Now we have noted this point in the final section of this paper (Line 17-20, Page 9).

Cryoconite mainly formed in the ablation zone on glacier, which indicates they are more active during summer season. Therefore, we only collected the cryoconite samples during summer time, as listed in Table 1. Now we have added this point in the text (Line 21-23, Page 3). The five research sites represent different geographic environments cross the Tibetan Plateau from north to south, which have been specifically described in Section 2.1. Regarding the spatial representativeness (Line 3-

7, Page 9), the glacier (i.e. BS) located in the southeast margin of Tibetan Plateau presents distinct pattern of the iron speciation (shown in Fig 4) and apportion of different constituents to the light absorption. The most plausible reason is that glacier is strongly impacted by anthropogenic emissions nearby, with the highest contents of black carbon and organic matters in cryoconites.

7. Section 3.3, Page 7, line 8: do you mean the effect of atmospheric aging on minerals? Please be more specific.

**Reply**: It is similar to the atmospheric aging but take place on the surface of glacier. Just as mentioned in line 4 -5 on page 7, goethite is favored in moist and cool conditions while hematite commonly occurs in warm and dry environment. Therefore, hematite may tend to transfer into goethite on the moist and cool cryoconite surface. Microorganisms may also play a crucial role in the degradation of rocks and minerals in terrestrial environment through changing the transformation rates, pathways, and even the end products (Fru et al., 2012). But more research is needed to test the hypothesis (Line 22-25, Page 7).

8. Page 8, lines 8-11: I guess this is basically your key conclusion and I would move it to Sect. 4. Also I suggest to add a brief discussion on the impact of the regional variability of iron oxides and their speciation, and the representativeness of your results compared to other regions of the world under the influence of other deserts with different mineralogical compositions.

**Reply**: We have moved the sentence (see Line 12-14, Page 9). And the spatial representativeness of our results was discussed (Line 3-7, Page 9).

[revised manuscript text omitted]

---

## Editor Decision (ED1)

Editor review

Thanks for your revisions. I agree with the reviewer that the manuscript needs English language corrections. I spent quite a bit of time going through your initial submission, correcting the English. However, much of the added wording after revisions contains grammatical issues. Please make corrections based on my specific comments below, in addition to the corrections suggested by the reviewer. What I've written below is how these sentences should read to be grammatically correct. Please make sure that these corrections accurately represent what you meant to communicate.

Page 1 line 17: Croyconite is a mixture of impurities and ice visually represented by dark colors present in the ablation zone of glaciers.

Page 3 lines 20-24: In the laboratory, the cryoconite samples were freeze-dried into powder. In this work, concentrations and fractions are referred to as dry cryoconite mass. Cryoconite mainly forms in the ablation zone on glaciers in summer season. Therefore, our measurements mainly reflect processes occurring in summer on the glacier surface.

Page 3 line 26: In detail, the sample was first transferred into a PTFE high

Page 3 line 32: This procedure was repeated to remove HF completely. You also need subscripts on HNO3 and H2O.

Page 3 line 33: to the vessel which was placed into the oven

Page 3 line 35: After treatment, each sample was measured by…

Page 5 line 25: at specific wavelengths

Page 5 line 27: mass absorption coefficient

Page 5 line 29: was calculated by dividing

Page 6 line 27: 7.3%

Page 7 line 35: terrestrial environments

Page 7 line 36: highest TOC content

Page 7 line 37: test this hypothesis

Page 9 line 19: (Remove beginning of this sentence). Four glaciers…

Page 9 line 20: similar patterns

Page 9 line 22: organic matter

Page 9 line 23: microbes in such environments may perturb the natural distribution

Page 9 line 29: the high mass concentration

Page 9 line 30: total absorption larger than BC.

Page 9 line 35: In future research,

---

## Author Response (AR2)

Dear Professor Alexander,

Thank you and the reviewer for the kindly suggestions. Now We have made the corrections accordingly and tried our best to improve the writing.

We really appreciate your efforts and help during the whole process.

Best regards,

Zhiyuan Cong